# CTSR: CONTROLLABLE FIDELITY-REALNESS TRADE-OFF DISTILLATION FOR REAL-WORLD IMAGE SUPER RESOLUTION

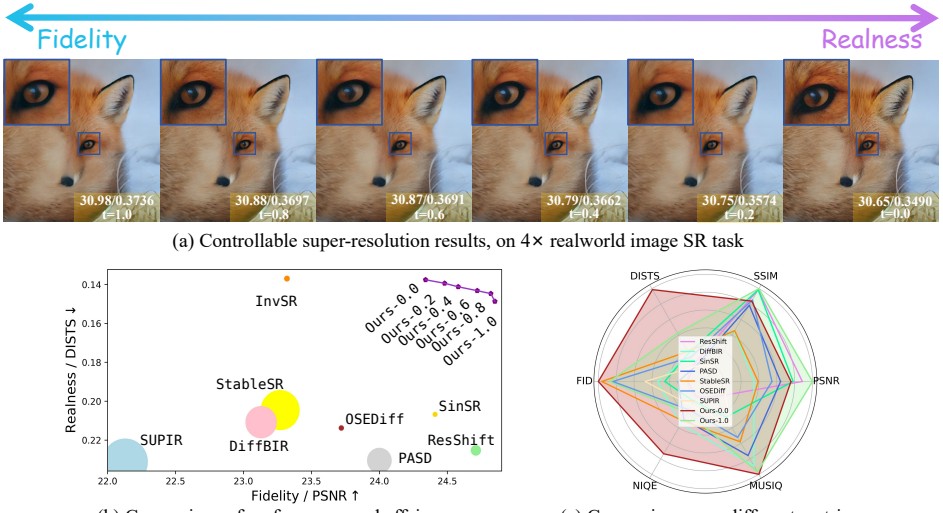

Figure 1: (a) Controllable trade-off of our proposed CTSR, which could be adjusted freely between better fidelity and realness. (b) Comparison of current state-of-the-art (SOTA) real-world image SR methods and CTSR on performance and efficiency. Larger bubble indicates longer inference time. The closer the bubble of a method is to the top-right corner of the figure, the better its performance in both fidelity and realness. For our controllable method, we select six different states and present their performance. The purple curve shows continuously adjusted trade-off points, all of which exhibit performance advantages. (c) Comparison on multiple metrics with current SOTA methods and CTSR. All results are done on DIV2K validation set, 4× SR with realworld degradation.

## ABSTRACT

Real-world image super-resolution is a critical image processing task, where two key evaluation criteria are the fidelity to the original image and the visual realness of the generated results. Although existing methods based on diffusion models excel in visual realness by leveraging strong priors, they often struggle to achieve an effective balance between fidelity and realness. In our preliminary experiments, we observe that a linear combination of multiple models outperforms individual models, motivating us to harness the strengths of different models for a more effective trade-off. Based on this insight, we propose a distillation-based approach that leverages the geometric decomposition of both fidelity and realness, alongside the performance advantages of multiple teacher models, to strike a more balanced trade-off. Furthermore, we explore the controllability of this trade-off, enabling a flexible and adjustable super-resolution process, which we call CTSR (Controllable Trade-off Super-Resolution). Experiments conducted on several real-world image super-resolution benchmarks demonstrate that our method surpasses existing state-of-the-art approaches, achieving superior performance across both fidelity and realness metrics.

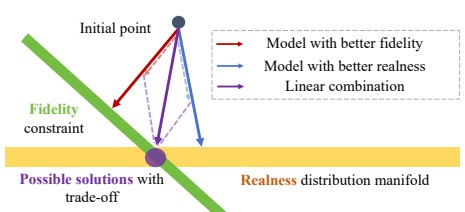

Figure 2: Illustration for vector decomposition in the image SR process. It shows the simple linear combination approach, which serves as the **motivation** of our proposed CTSR.

Table 1: Results of the linear combination on RealSR Cai et al. (2019) Nikon sub-testset. $\alpha$ is multiplied with ResShift Yue et al. (2023), and $(1-\alpha)$ with OSEDiff Wu et al. (2024a). By adding SR results from two models, the performance for both fidelity and realness is improved. Best and second-best results shown in **red** and blue.

| Settings | PSNR↑ | LPIPS↓ | Inference time (s) |
|---|---|---|---|
| $\alpha = 0$ | 24.54 | 0.3575 | 0.7546 |
| $\alpha = 0.2$ | 24.84 | 0.3525 | 0.9196 |
| $\alpha = 0.4$ | 25.25 | 0.3633 | 0.9196 |
| $\alpha = 0.6$ | 25.34 | 0.3742 | 0.9196 |
| $\alpha = 0.8$ | 25.10 | 0.3857 | 0.9196 |
| $\alpha = 1.0$ | 24.88 | 0.3915 | **0.1791** |
| Ours | **25.45** | **0.3411** | **0.1791** |

# 1 INTRODUCTION

Image restoration, particularly image super-resolution (SR), is both a critical and challenging task in image processing. Early research Yang et al. (2010); Kim & Kwon (2010); Wang et al. (2015) typically focused on fixed degradation operators, such as downsampling and blur kernels, modeled as $y = Ax + n$, where $x$ represents the original image, $A$ is the fixed degradation operator, $n$ is random noise, and $y$ is the degraded result. As the field has advanced, more recent work has shifted its focus to real-world degradation scenarios, where $A$ turns to a complex and random combination of various degradations, with unknown degradation types and parameters. The evaluation of image super-resolution is mainly based on two metrics: fidelity, which measures the consistency between the super-resolved image and the degraded image, and realness, which assesses how well the super-resolved image conforms to the prior distribution of natural images, as well as its visual quality Mentzer et al. (2020); Zhou & Wang (2022); Zhang et al. (2022). The early methods primarily used architectures based on GAN Goodfellow et al. (2014) and MSE, trained on pairs of original and degraded images Dong et al. (2015); Liang et al. (2021); Wang et al. (2018); Guo et al. (2022). These approaches excelled in achieving good fidelity in super-resolved results but often suffered from over-smoothing and detail loss Chen et al. (2024). The introduction of diffusion models brought powerful visual priors to the SR task, significantly improving the realness and visual quality of super-resolved images. However, these models frequently struggle with maintaining consistency between the super-resolved and degraded images. Achieving a satisfactory balance between fidelity and realness remains a challenge, with most methods failing to strike an effective trade-off.

The core challenge of real-world image super-resolution lies in addressing an inherent multi-objective optimization problem. Given a low-resolution observation $y$, the goal is to recover a high-resolution $\hat{x}$ that simultaneously satisfies two conflicting criteria: *fidelity*, where $\hat{x}$ should be close to the ground-truth image $x_{GT}$, typically quantified by minimizing a distortion measure $D(\hat{x}, x_{GT})$ such as MSE, and *realness*, where $\hat{x}$ should appear natural and conform to the statistical distribution of real-world images, $p_{data}(x)$. Foundational work in image restoration has established that these objectives are bound by an unavoidable "Perception-Distortion (P-D) Tradeoff". No algorithm can simultaneously achieve zero distortion and perfect perceptual quality. All optimally achievable solutions form a **Pareto Front**, which manifests as a convex curve in the P-D plane.

Our initial exploratory experiments in Tab. 1 revealed an interesting phenomenon: a linear combination of outputs from a high-fidelity model $G_f$ and a high-realness model $G_r$, denoted as $\hat{x}_c = \alpha \hat{x}_f + (1-\alpha)\hat{x}_r$, could surpass either individual model on certain metrics. However, the P-D tradeoff theory reveals the fundamental limitation of this naive linear approach. Assuming $\hat{x}_f$ and $\hat{x}_r$ correspond to two distinct points $(D_f, P_f)$ and $(D_r, P_r)$ on the Pareto-optimal curve, any linear interpolation $\hat{x}_c$ in the image space will almost certainly yield a point $(D_c, P_c)$ that lies *below* the chord connecting the two *initial* points, and thus within the sub-optimal region enclosed by the convex Pareto front. This implies that for any solution obtained via linear combination, a theoretically superior solution $x^*$ exists on the Pareto curve that is strictly better in at least one metric.

Therefore, our initial observation should not be interpreted as a viable solution, but rather as a crucial insight: an optimal trade-off point exists in the solution space between these two experts, but it does not lie on the linear path connecting them. This leads to the core motivation of our work: **Can we design a framework to train a single, efficient student model $\mathcal{S}$ that learns to operate directly on**

**the Pareto-optimal curve of the P-D tradeoff, rather than interpolating on a sub-optimal linear path?** To address this, we propose CTSR, a controllable trade-off real-world image super-resolution method based on fidelity-realness distillation. The core idea is to leverage high-fidelity and high-realness teacher models not for their outputs, but as "expert guides" providing gradient signals from different optimization directions Chung et al. (2022); Soh et al. (2019). This guides the student model to discover a new, superior operating point on the Pareto front. Furthermore, to achieve a continuous and controllable trade-off, we further distill the model using a flow-matching-inspired technique Lipman et al. (2024); Zhu et al. (2024c); Fischer et al. (2023), enabling it to traverse the learned optimal path and freely adjust between fidelity and realness. As demonstrated in Fig. 1, our CTSR enables fine-grained control over the SR results. To summarize, our contributions are three-fold:

❑ We propose a real-world image super-resolution method based on fidelity-realness distillation, effectively achieving a trade-off between fidelity and realness.

❑ We further introduce a continuous and controllable trade-off approach through another distillation process, enabling the model to freely adjust the balance between fidelity and realness, thus providing practical user flexibility and advancing the optimization of image SR tasks.

❑ Experiments on real-world image SR benchmarks demonstrate the superior performance of our proposed CTSR method, along with efficient inference sampling steps and reduced trainable parameter count.

## 2 RELATED WORK

**Diffusion-based SR with Fixed Degradation** Earlier works on image SR Lin & Shum (2004); Farsiu et al. (2004); Elad & Aharon (2006); Elad & Feuer (1997); Zeyde et al. (2010); Jiji et al. (2004; 2007) usually use gradient-based methods to optimize image matrix Sun et al. (2008; 2010), which inspires the following diffusion-based approaches to use LR input as guidance for diffusion sampling iteration. As diffusion models have developed, their strong visual priors have also been applied to image super-resolution tasks. SR3 Saharia et al. (2022) first proposes a diffusion model for the SR task, which uses LR input as a condition of diffusion sampling, thus requiring training for the UNet. Further methods like DDRM Kawar et al. (2022), DDNM Wang et al. (2023b) and DPS Chung et al. (2023) use classifier-free guidance Ho & Salimans (2022), which takes LR input as the guidance of original diffusion sampling; thus, these methods are training-free. However, all of these methods are on a fixed degradation setting, where the degradation type and parameters are known.

**Diffusion-based SR with Real-world Settings** As these training-free methods use gradient guidance to correct the diffusion sampling process, methods such as DiffBIR Xinqi et al. (2024) and GDP Fei et al. (2023) try to leverage the gradient to update the parameters of the degradation operator, and in this case the degradation parameters are unknown. The current diffusion-based image SR methods focus mainly on the real-world scenario, where the degradation is unknown and complex Wang et al. (2024a); Xie et al. (2024); Wu et al. (2024b); Wang et al. (2024b); Wu et al. (2024a); Yue et al. (2023); Yang et al. (2024); Yu et al. (2024). StableSR Wang et al. (2024a) proposes an SR method based on Stable Diffusion Rombach et al. (2022), using an adapter to introduce the LR guidance for diffusion sampling. However, such an approach requires multiple steps to obtain the SR result, which is time-consuming. ResShift Yue et al. (2023) designs a special sampling, accelerating the overall sampling in 15 steps. Currently, some methods try to distill the diffusion-based methods into one step, including AddSR Xie et al. (2024), SinSR Wang et al. (2024b) and OSEDiff Yu et al. (2024). Some papers also explore the controllability of diffusion-based SR, including PiSA-SR Sun et al. (2025) and OFTSR Zhu et al. (2024c).

## 3 METHOD

### 3.1 MOTIVATION

In diffusion-based methods, some approaches excel in fidelity, such as ResShift Yue et al. (2023) and SinSR Wang et al. (2024b), while others prioritize realness metrics, like OSEDiff Wu et al. (2024a)

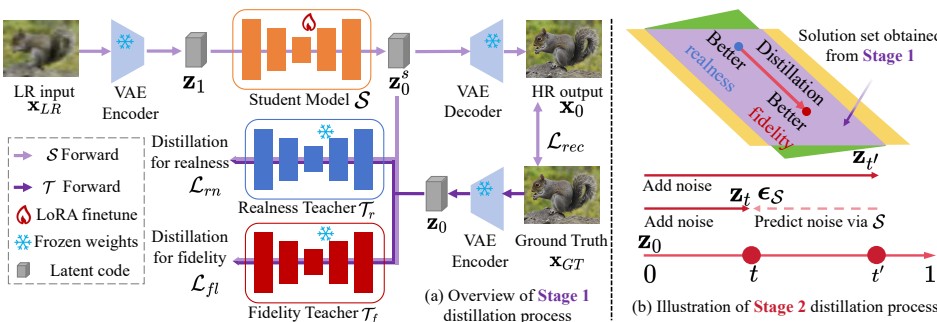

Figure 3: Illustration of our proposed CTSR. (a) At the first stage, we distill student model via two teacher models, one with better fidelity performance, and one with better realness performance. (b) At the second stage, we distill model obtailed from first stage, to a continuous mapping to SR results with different trade-offs between fidelity and realness.

and StableSR Wang et al. (2024a). Combining the strengths of these methods can facilitate an effective trade-off between the two. One straightforward approach is to linearly combine the super-resolved outputs of different models. For example, by multiplying the image tensor of ResShift by $\alpha$ and OSEDiff by $(1-\alpha)$, and then summing them, both fidelity and realness metrics can be improved by adjusting the coefficients. We validate this on the Nikon test subset of RealSR Cai et al. (2019), with the results shown in Tab. 1. We further interpret this linear combination method as the sum of vectors corresponding to different SR methods in the image space, as illustrated in Fig. 2.

However, the performance of the above linear combination method is limited and its inference speed is slower because of the need to run two models. To address these issues and enhance the model's representation capability, we extend it to a more general framework. Inspired by the success of knowledge distillation Liu et al. (2020); Shao et al. (2023) in image SR Hui et al. (2019); Zhang et al. (2021b; 2024b); Zhu et al. (2024a), we distill the model output to the intersection of consistency constraints and high-quality image distribution manifolds, striking a trade-off of fidelity and realness. To further enable controllability of the trade-off between fidelity and realness, we distill the diffusion sampling process of the model into a transformation from realness to fidelity, allowing for a flexible, controllable adjustment between the two. As a result, users can freely adjust these two properties according to their preferences in practical scenarios.

## 3.2 OVERVIEW

Our model is an one-step diffusion-based SR approach finetuned from OSEDiff Wu et al. (2024a). The training scheme consists of two stages. In the first stage, as shown in Fig. 3(a), we select an SR model with good realness as the student model $\mathcal{S}$. This model is distilled via LoRA Hu et al. (2022) using two teacher models: one with high fidelity (denoted as $\mathcal{T}_f$) and another with good realness (denoted as $\mathcal{T}_r$). The teacher model $\mathcal{T}_f$ guides the student model $\mathcal{S}$ with gradient directions for fidelity, while $\mathcal{T}_r$ ensures that the student model retains its original generative capability. As a result, the super-resolution process of the model receives gradient corrections in the fidelity direction, and converges to the intersection of the fidelity constraint and the realness distribution manifold.

In the second stage, as shown in Fig. 3(b), we further distill $\mathcal{S}$ within the solution set obtained from the first stage. Since the diffusion model can be viewed as a distribution transformation mapping from the initial input to the final output, we set the starting point as the super-resolved result from the first stage, with the target transformation being the solution with better fidelity within the solution set. This distribution transformation is achieved through distillation. As the time step $t$ of the diffusion model is continuous, we can controllably select the appropriate trade-off state, allowing us to achieve better and more diverse super-resolution results. An illustration of our proposed CTSR is shown in Fig. 3.

## 3.3 STAGE 1: DISTILLATION VIA DUAL-TEACHER LEARNING

Motivated by the insight in Sec. 3.1, we propose a distillation-based method, where two super-resolution models with good fidelity $\mathcal{T}_f$ and realness $\mathcal{T}_r$, are used to distill the original model $\mathcal{S}$. Our training objective consists of two components:

**Reconstruction Loss**. The output of the student model should be consistent with the original model in terms of both consistency and visual quality. We choose $L_2$ loss and LPIPS loss as the reconstruction loss terms:

$$\mathcal{L}_{rec} = \lambda_{l2}||\mathcal{S}(\mathbf{x}_{LR}) - \mathbf{x}_{GT}||_2^2 + \lambda_{lp}\ell(\mathcal{S}(\mathbf{x}_{LR}), \mathbf{x}_{GT}) \tag{1}$$

, where $\mathbf{x}_{LR}$ is input LR image, $\mathbf{x}_{GT}$ is ground-truth image, $\ell$ is LPIPS loss, $\lambda_{l2}$ and $\lambda_{lp}$ are balancing hyper-parameters.

**Dual Teacher Distillation Loss**. For ease of implementation, we use the same model for both the realness teacher $\mathcal{T}_r$ and the student model $\mathcal{S}$. This allows us to divide the distillation process into two parts: (1) The fidelity teacher model $\mathcal{T}_f$ guides the gradients of $\mathcal{S}$, adjusting its output distribution in a more faithful direction. (2) The realness teacher model $\mathcal{T}_r$ regulates the student model, ensuring that the directional correction in (1) does not deviate from the manifold of the true image distribution achieved by $\mathcal{T}_r$. The specific formula for $\mathcal{L}_{fl}$ is as follows:

$$\mathcal{L}_{fl} = ||\boldsymbol{\epsilon}_{\mathcal{T}_f}(\mathbf{z}_t^s, t, c) - \boldsymbol{\epsilon}_{\mathcal{S}}(\mathbf{z}_t^s, t, c)||_2^2 \\ + \gamma_{time}||\boldsymbol{\epsilon}_{\mathcal{T}_f}(\mathbf{z}_t^s, t, c) - \boldsymbol{\epsilon}_{\mathcal{T}_f}(\mathbf{z}_t, t, c)||_2^2, \tag{2}$$

where $\boldsymbol{\epsilon}_{\mathcal{T}_f}$ and $\boldsymbol{\epsilon}_{\mathcal{S}}$ represent the denoising UNet of $\mathcal{T}_f$ and $\mathcal{S}$, respectively; $c$ is the prompt embedding; $\mathbf{z}_t$ and $\mathbf{z}_t^s$ are the latent codes of ground-truth $\mathbf{x}_{GT}$ and the student model's SR result $\mathbf{x}_0$, obtained via VAE encoder $\mathcal{E}$, each added with the noise at timestep $t$ in the forward process of the diffusion model; $\gamma_{time}$ is the hyperparameter to balance the two terms. The first term $\boldsymbol{\epsilon}_{\mathcal{T}_f}(\mathbf{z}_t^s, t, c) - \boldsymbol{\epsilon}_{\mathcal{S}}(\mathbf{z}_t^s, t, c)$ aligns the output of $\mathcal{S}$ with the teacher model $\mathcal{T}_f$, enabling the student model to learn the distribution information from the teacher. The second term, $\boldsymbol{\epsilon}_{\mathcal{T}_f}(\mathbf{z}_t^s, t, c) - \boldsymbol{\epsilon}_{\mathcal{T}_f}(\mathbf{z}_t, t, c)$, leverages the teacher model's prior to align the SR result $\mathbf{x}_0$ with $\mathbf{x}_{GT}$. Since alignment in the second term is achieved by adding noise to the latent codes of $\mathbf{x}_0$ and $\mathbf{x}_{GT}$ separately, and calculating the difference in the predicted noise of $\mathcal{T}_f$, it reflects the distributional difference between them in the image space. As a result, compared to directly using $L_2$ loss, this approach better captures the distributional differences between the student model and the ground truth, avoiding issues like over-smoothing and loss of detail typically introduced by $L_2$ loss, while preserving the semantic details of the original image. We show the detailed calculation process of $\mathcal{L}_{fl}$ in Fig. 6 of **Appendix**.

This design is similarly applied for the distillation of $\mathcal{T}_r$:

$$\mathcal{L}_{rn} = ||\boldsymbol{\epsilon}_{\mathcal{T}_r}(\mathbf{z}_t^s, t, c) - \boldsymbol{\epsilon}_{\mathcal{S}}(\mathbf{z}_t^s, t, c)||_2^2 \\ + \gamma_{time}||\boldsymbol{\epsilon}_{\mathcal{T}_r}(\mathbf{z}_t^s, t, c) - \boldsymbol{\epsilon}_{\mathcal{T}_r}(\mathbf{z}_t, t, c)||_2^2, \tag{3}$$

By combining these losses, the student model $\mathcal{S}$ can achieve improved fidelity without sacrificing its original performance. As a result, the linear combination method discussed in Sec. 3.1 is extended to a more general approach, where the student's convergence direction evolves from a simple vector sum to a more precise optimal solution direction. This distillation mechanism is inspired by the SDS Poole et al. (2022) and VSD Wang et al. (2023c); Dong et al. (2024) losses, which regulate the student model using both the teacher model and the ground truth.

The loss function for distillation in the first stage is:

$$\mathcal{L}_{s1} = \mathcal{L}_{rec} + \lambda_{rn}\mathcal{L}_{rn} + \lambda_{fl}\mathcal{L}_{fl}, \tag{4}$$

where $\lambda_{rn}$ and $\lambda_{rn}$ are balancing weights.

In short, our proposed distillation method guides the student model $\mathcal{S}$ toward the intersection of the fidelity constraint and the realness distribution. The distilled SR model then serves as the teacher model in the following second stage, providing SR solutions with fidelity-realness trade-off.

### 3.4 STAGE 2: DISTILLATION FOR CONTROLLABILITY

The first stage of our method yields a student model, which we now denote as $\mathcal{S}_1$, that is optimized to produce a single, high-quality solution on the Perception-Distortion (P-D) Pareto front. The goal of our second stage is to endow this model with controllability, allowing a user to navigate along this optimal front. To achieve this in a principled manner, we reformulate this stage based on the

Table 2: Quantitative comparison of the state-of-the-art methods with superior performance on *fidelity*. $t$ is the timestep set in ours CTSR. The best and second-best results of each metric are highlighted in **red** and blue. M-IQ for MUSIQ, M-IQA for MANIQA and C-IQA for CLIPIQA.

| Datasets | Method | PSNR↑ | SSIM↑ | LPIPS↓ | DISTS↓ | FID↓ | NIQE↓ | M-IQ↑ | M-IQA↑ | C-IQA↑ |
|---|---|---|---|---|---|---|---|---|---|---|
| **DRealSR** | RealESRGAN Wang et al. (2021) | 28.62 | 0.8052 | 0.5428 | 0.2374 | 171.79 | 7.8675 | 54.26 | 0.5202 | 0.4515 |
| | ResShift Yue et al. (2023) | **28.69** | 0.7874 | **0.3525** | 0.2541 | 176.77 | 7.8762 | 52.40 | 0.4756 | 0.5413 |
| | SinSR Wang et al. (2024b) | 28.38 | 0.7497 | 0.3669 | 0.2484 | 172.72 | **6.9606** | 55.03 | 0.4904 | 0.6412 |
| | CTSR (t=0.8) (ours) | 28.47 | **0.8056** | 0.3561 | 0.2369 | **161.24** | 7.8462 | 58.76 | 0.5453 | 0.6745 |
| **RealSR** | RealESRGAN Wang et al. (2021) | 25.69 | **0.7614** | 0.3266 | 0.1646 | 168.02 | **4.0146** | 60.36 | 0.3934 | 0.4495 |
| | ResShift Yue et al. (2023) | **26.39** | 0.7567 | **0.3158** | 0.2432 | 149.59 | 6.8746 | 60.22 | 0.5419 | 0.5496 |
| | SinSR Wang et al. (2024b) | 26.27 | 0.7351 | 0.3217 | 0.2341 | 137.59 | 6.2964 | 60.76 | 0.5418 | 0.6163 |
| | CTSR (t=0.2) (ours) | 26.29 | 0.7211 | 0.3210 | **0.1620** | **127.67** | 4.2979 | 66.84 | 0.6314 | 0.6435 |
| **DIV2K-Val** | RealESRGAN Wang et al. (2021) | 24.29 | **0.6372** | 0.3570 | 0.1621 | 46.31 | **3.4591** | 61.05 | 0.3830 | 0.5276 |
| | ResShift Yue et al. (2023) | **24.71** | 0.6234 | 0.3473 | 0.2253 | 42.01 | 6.3615 | 60.63 | 0.5283 | 0.5962 |
| | SinSR Wang et al. (2024b) | 24.41 | 0.6018 | 0.3262 | 0.2068 | 35.55 | 5.9981 | 62.95 | 0.5430 | 0.6501 |
| | CTSR (t=0.2) (ours) | 24.45 | 0.6098 | 0.3384 | **0.1394** | 24.75 | 3.6803 | 69.25 | 0.5826 | 0.6726 |

Table 3: Quantitative comparison of methods with better performance on *realness*. $t$ is the timestep of set in our CTSR. The best and second-best results of each metric are highlighted in **red** and blue.

| Datasets | Method | PSNR↑ | SSIM↑ | LPIPS↓ | DISTS↓ | FID↓ | NIQE↓ | MUSIQ↑ | MANIQA↑ | CLIPIQA↑ |
|---|---|---|---|---|---|---|---|---|---|---|
| **DRealSR** | StableSR Wang et al. (2024a) | **28.04** | 0.7454 | 0.3279 | 0.2272 | 144.15 | 6.5999 | 58.53 | 0.5603 | 0.6250 |
| | DiffBIR Xinqi et al. (2024) | 25.93 | 0.6525 | 0.4518 | 0.2761 | 177.04 | **6.2324** | **65.66** | 0.6296 | 0.6860 |
| | SUPIR Yu et al. (2024) | 25.09 | 0.6460 | 0.4243 | 0.2795 | 169.48 | 7.3918 | 58.79 | 0.5471 | 0.6749 |
| | PASD Yang et al. (2024) | 27.79 | 0.7495 | 0.3579 | 0.2524 | 171.03 | 6.7661 | 63.23 | 0.5919 | 0.6242 |
| | InvSR Yue et al. (2024) | 26.75 | 0.6870 | 0.4178 | 0.2144 | 142.98 | 6.7030 | 63.92 | 0.5439 | 0.6791 |
| | OSEDiff Wu et al. (2024a) | 27.35 | 0.7610 | **0.3177** | 0.2365 | 141.93 | 7.3053 | 63.56 | 0.5763 | 0.7053 |
| | CTSR (t=0.0) (ours) | 27.38 | **0.7767** | 0.3423 | **0.1937** | 142.52 | 6.6438 | 64.70 | **0.6412** | 0.7060 |
| **RealSR** | StableSR Wang et al. (2024a) | 24.62 | 0.7041 | 0.3070 | 0.2156 | 128.54 | 5.7817 | 65.48 | 0.6223 | 0.6198 |
| | DiffBIR Xinqi et al. (2024) | 24.24 | 0.6650 | 0.3469 | 0.2300 | 134.56 | 5.4932 | **68.35** | 0.6544 | 0.6961 |
| | SUPIR Yu et al. (2024) | 23.65 | 0.6620 | 0.3541 | 0.2488 | 130.38 | 6.1099 | 62.09 | 0.5780 | 0.6707 |
| | PASD Yang et al. (2024) | 25.68 | 0.7273 | 0.3144 | 0.2304 | 134.18 | 5.7616 | 68.33 | 0.6323 | 0.5783 |
| | InvSR Yue et al. (2024) | 24.50 | 0.7262 | **0.2872** | 0.1624 | 148.16 | 4.2189 | 67.45 | **0.6636** | 0.6918 |
| | OSEDiff Wu et al. (2024a) | 23.94 | 0.6736 | 0.3172 | 0.2363 | 125.93 | 6.3822 | 67.52 | 0.6187 | **0.7001** |
| | CTSR (t=0.0) (ours) | **25.70** | 0.6962 | 0.3058 | **0.1530** | 121.30 | **4.0662** | 67.94 | 0.6367 | 0.6495 |
| **DIV2K-Val** | StableSR Wang et al. (2024a) | 23.27 | 0.5722 | 0.3111 | 0.2046 | 24.95 | 4.7737 | 65.78 | 0.6164 | 0.6753 |
| | DiffBIR Xinqi et al. (2024) | 23.13 | 0.5717 | 0.3469 | 0.2108 | 33.93 | 4.6056 | 68.54 | **0.6360** | 0.7125 |
| | SUPIR Yu et al. (2024) | 22.13 | 0.5279 | 0.3919 | 0.2312 | 31.40 | 5.6767 | 63.86 | 0.5903 | 0.7146 |
| | PASD Yang et al. (2024) | 24.00 | 0.6041 | 0.3779 | 0.2305 | 39.12 | 4.8587 | 67.36 | 0.6121 | 0.6327 |
| | InvSR Yue et al. (2024) | 23.32 | 0.5901 | 0.3657 | **0.1370** | 28.85 | 3.0567 | 68.97 | 0.6122 | **0.7198** |
| | OSEDiff Wu et al. (2024a) | 23.72 | **0.6109** | **0.3058** | 0.2138 | 26.34 | 5.3903 | 65.27 | 0.5838 | 0.6558 |
| | CTSR (t=0.0) (ours) | 24.34 | 0.6093 | 0.3377 | 0.1377 | 24.56 | 3.5455 | 69.52 | 0.5894 | 0.6741 |

Rectified Flow framework (Lipman et al., 2024), correcting the mathematical inconsistencies in our initial approach. Rectified Flow provides a powerful and theoretically sound method for learning a direct, efficient mapping between two data distributions, $\pi_0$ and $\pi_1$. It models this transformation as an Ordinary Differential Equation (ODE), $d_{\mathbf{z}_t} = v(\mathbf{z}_t, t)$, where $v(\mathbf{z}_t, t)$ is a velocity vector field learned by a neural network. The core insight of Rectified Flow is to train this velocity field to transport samples along straight-line paths, simplifying both training and inference. We adapt this framework to our specific task by defining the source and target distributions for the desired P-D trajectory:

**Source Distribution** $\pi_0$: This is the distribution of high-quality SR images generated by Stage 1 model, $\mathbf{S}_1$. For any given LR input $\mathbf{x}_{LR}$, a sample from this distribution is $\mathbf{x}_0 = \mathcal{S}_1(\mathbf{x}_{LR})$. This represents our optimal starting point on the Pareto front, corresponding to $t = 0$.

**Target Distribution** $\pi_1$: This is the distribution of high-fidelity SR images generated by the fidelity teacher model, $\mathcal{T}_f$. For the same input $\mathbf{x}_{LR}$, a sample is $\mathbf{x}_1 = \mathcal{T}_f(\mathbf{x}_{LR})$. This defines the endpoint of our trajectory, corresponding to $t = 1$.

Our objective is to learn a velocity field $v_{\mathcal{S}}$ that can transport an image from the distribution $\pi_0$ to $\pi_1$ in a single conceptual step. We operate in the latent space of the VAE. Let $\mathbf{z}_0 = \mathcal{E}(\mathbf{x}_0)$ and $\mathbf{z}_1 = \mathcal{E}(\mathbf{x}_1)$ be the latent representations of the source and target images, where $\mathcal{E}$ is the VAE encoder. The straight-line path connecting these points is parameterized as $\mathbf{z}_t = (1 - t)\mathbf{z}_0 + t\mathbf{z}_1$ for $t \in [0, 1]$. The target velocity vector along this path is constant and given by the simple difference $v_{target} = \frac{d\mathbf{z}_t}{dt} = \mathbf{z}_1 - \mathbf{z}_0$. The training objective for our student model $\mathcal{S}$ in this stage is to learn a velocity predictor $v_{\mathcal{S}}$ that accurately estimates this target velocity for any point $\mathbf{z}_t$ along the path. This is formulated as a simple mean squared error loss:

$$\mathcal{L}_{s2} = \mathbb{E}_{\mathbf{x}_{LR}, t \sim [0,1]} ||(\mathbf{z}_1 - \mathbf{z}_0) - v_{\mathbf{S}}((1 - t)\mathbf{z}_0 + t\mathbf{z}_1, t, c)||_2^2 \tag{5}$$

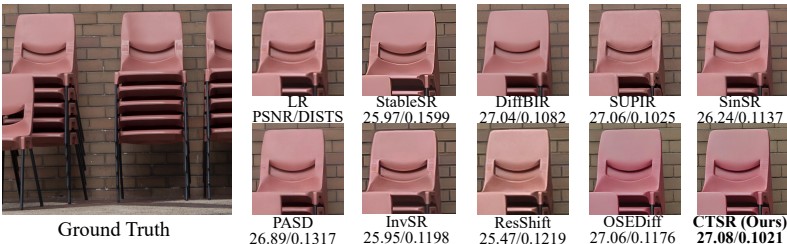

Figure 4: Visualized results of evaluation on the RealSR testset, with our proposed CTSR ($t = 0.0$) and compared methods.

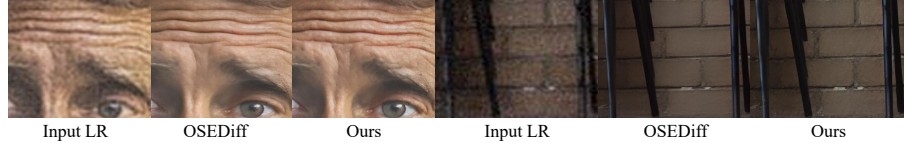

Figure 5: Detailed comparison on RealSR validation set, zoom in for more details.

where $c$ represents prompt embeddings. This objective directly trains the student network to predict the direction of the full trajectory from the balanced solution to the high-fidelity solution.

This new formulation provides a clear and direct mechanism for control at inference. The parameter $t$ now represents the desired position along the learned trajectory. Given input $\mathbf{x}_{LR}$, we first compute the start-point latent $\mathbf{z}_0 = \mathcal{E}(\mathcal{S}_1(\mathbf{x}_{LR}))$. To generate a super-resolved image at a specific trade-off level $t_{infer}$, we approximate the solution to the learned ODE with a single Euler step:

$$\mathbf{z}_{out}(t_{infer}) = \mathbf{z}_0 + t_{infer} \cdot v_{\mathcal{S}}(\mathbf{z}_0, 0, c) \tag{6}$$

The final image is then produced by the VAE decoder: $\hat{\mathbf{x}}_t = \mathcal{D}(\mathbf{z}_{out}(t_{infer}))$. As $t_{infer}$ increases towards 1, the output is progressively shifted along the learned vector field towards the high-fidelity domain. This provides an efficient, one-step, and theoretically grounded method for achieving a continuous and controllable fidelity-realness trade-off.

## 4 EXPERIMENTS

### 4.1 SETTINGS

**Datasets** We merge the training sets from DIV2K Agustsson & Timofte (2017), LSDIR Li et al. (2023), DRealSR Wei et al. (2020), ImageNet Deng et al. (2009), and RealSR Cai et al. (2019) as our training dataset, and evaluate our method on the validation sets of DIV2K, DRealSR, and RealSR. The degraded images are generated using the real-world degradation operator from RealESR-GAN Wang et al. (2021). For the SR process, we first up-sample the degraded images in the scaling factor of $\times 4$, then input them into our proposed SR framework, so the size of the degraded input and the obtained output are matched. The task real-world image super-resolution here is not limited to up-sampling image to a larger size, but also includes other restoration process, like removal of alias, blur, and noise, to improve the visual quality of input image.

**Evaluation Metrics** We assess both fidelity and realness for evaluation. For fidelity, we use PSNR and SSIM Wang et al. (2004); for realness, we use LPIPS Zhang et al. (2018), DISTS Ding et al. (2020), and FID Heusel et al. (2017), which require reference images, and NIQE Zhang et al. (2015), MUSIQ Ke et al. (2021), CLIPIQA Wang et al. (2023a), and MANIQA Yang et al. (2022), which are reference-free. LPIPS uses VGG Simonyan & Zisserman (2014) weights following Dong et al. (2024), and MANIQA uses PIPAL Jinjin et al. (2020) weights by default.

**Implementation Details** For the teacher model selection, we choose OSEDiff Wu et al. (2024a) as $\mathcal{T}_r$, due to its advantage in realness, and ResShift Yue et al. (2023) as $\mathcal{T}_r$, due to its better fidelity performance. The pretrained version of Stable Diffusion Rombach et al. (2022) used is 2.1-base. The default image input size for the models is $512 \times 512$. All images are processed at their original size, and for images larger than $512 \times 512$, we use patch splitting and apply VAE tiling to avoid block artifacts. In both the first and second stages of training, we use the AdamW Loshchilov & Hutter

Table 4: Ablation of training with different teachers, and without dual teacher distillation loss. Best and second-best results are shown in **red** and blue. C-IQA and M-IQA are short for CLIPIQA and MANIQA.

| Teacher $\mathcal{T}_{fl}$ | PSNR↑ | SSIM↑ | LPIPS↓ | C-IQA↑ | M-IQA↑ |
|---|---|---|---|---|---|
| w/o distill | **26.71** | 0.6743 | 0.4552 | 0.5439 | 0.5775 |
| SinSR | 25.71 | 0.6734 | 0.3552 | 0.6036 | 0.6065 |
| ResShift (Ours) | 25.70 | **0.6962** | **0.3058** | **0.6495** | **0.6367** |

Table 5: Our distillation applied in low-light enhancement task evaluated on LOL-v2-syn Chen et al. (2018) testset, which brings fidelity preservation and realness improvement. "Para." is short for parameters. Best results in **red**.

| Method | PSNR↑ | SSIM↑ | LPIPS↓ | Para. (M) ↓ |
|---|---|---|---|---|
| GSAD Hou et al. (2023) | 28.67 | 0.9444 | 0.0487 | **17.17** |
| Reti-Diff He et al. (2023) | 27.53 | 0.9512 | 0.0349 | 26.11 |
| GSAD (Distilled) | **28.69** | 0.9507 | **0.0336** | **17.17** |

(2017) optimizer with $\beta_1$=0.9, $\beta_2$=0.999, and a learning rate of 5e-5, with 20,000 training steps in the first stage and 50,000 in the second stage. The batch size is set to 1. Distillation in both stages is performed using LoRA Hu et al. (2022) fine-tuning, with a rank of 4. In the inference process, the prompt is obtained from a pre-trained RAM Zhang et al. (2024a) module, following current state-of-the-art methods Wu et al. (2024a); Sun et al. (2025). For the loss balancing coefficients in $\mathcal{L}_{s1}$, $\lambda_{rn}$ is set to 1, $\lambda_{fl}$ to 2, and $\gamma_{time}$ to 5.5. In $\mathcal{L}_{rec}$, $\lambda_{l2}$ and $\lambda_{lp}$ are set to 1 and 2 respectively. For the timestep $t$ in our CTSR, we set it as 0.0 for comparison with methods with better *realness*, and set $t$ to values greater than 0 for methods with better *fidelity*. Settings of compared methods are set as their default choice, and the final timestep $t$ of diffusion-based methods are 0. All experiments are conducted on SR task with a scaling factor of 4, using an NVIDIA A6000 GPU.

## 4.2 COMPARISON WITH STATE-OF-THE-ARTS

**Comparison Methods**. We select methods for comparison based on two performance metrics: fidelity and realness, and group them accordingly. For fidelity, we choose ResShift Yue et al. (2023), SinSR Wang et al. (2024b), and RealESRGAN Wang et al. (2021); for realness, we select StableSR Wang et al. (2024a), DiffBIR Xinqi et al. (2024), SUPIR Yu et al. (2024), SinSR Wang et al. (2024b), PASD Yang et al. (2024), InvSR Yue et al. (2024), and OSEDiff Wu et al. (2024a).

Table 6: Ablation for $\lambda_{rn}$, $\lambda_{fl}$ and $\lambda_{time}$. It is shown that our choice (in **bold**) leads to a better trade-off for both fidelity and realness. Best and second-best results shown in **red** and blue.

| $\lambda_{rn}$ | PSNR↑ | LPIPS↓ | $\lambda_{fl}$ | PSNR↑ | LPIPS↓ | $\gamma_{time}$ | PSNR↑ | LPIPS↓ |
|---|---|---|---|---|---|---|---|---|
| 0.6 | 25.07 | 0.3487 | 1.6 | 25.81 | 0.3377 | 4.5 | 25.08 | 0.3481 |
| 0.8 | 24.81 | 0.3185 | 1.8 | 25.62 | 0.3365 | 5.0 | 25.60 | 0.3166 |
| **1.0** | **25.70** | **0.3058** | **2.0** | **25.70** | **0.3058** | **5.5** | 25.70 | **0.3058** |
| 1.2 | 25.66 | 0.3376 | 2.2 | 25.44 | 0.3149 | 6.0 | 24.82 | 0.3212 |
| 1.4 | 25.62 | 0.3317 | 2.4 | 25.19 | 0.3226 | 6.5 | **27.07** | 0.3490 |

Table 7: Results of the controllable trade-off with adjustable properties implemented via timestep $t$, on DIV2K validation set. Best and second-best results in **red** and blue.

| Timestep $t$ | PSNR↑ | LPIPS↓ | NIQE↓ | MUSIQ↑ |
|---|---|---|---|---|
| 0.0 | 24.34 | **0.3377** | **3.5455** | **69.52** |
| 0.2 | 24.45 | 0.3384 | 3.6803 | 69.25 |
| 0.4 | 24.58 | 0.3397 | 3.8114 | 69.00 |
| 0.6 | 24.72 | 0.3409 | 3.9368 | 68.60 |
| 0.8 | 24.82 | 0.3423 | 4.0234 | 68.25 |
| 1.0 | **24.85** | 0.3437 | 4.0438 | 67.96 |

Table 8: Comparison of computational complexity and number of parameters across diffusion-based methods. Best and second-best results are shown in **red** and blue. Numbers in parentheses after method name is diffusion sampling steps. "Infer." is short for inference time (seconds), and "Para." for *trainable* parameters (M).

|  | StableSR(200) | DiffBIR(50) | SUPIR(50) | PASD(20) | ResShift(15) | InvSR(1) | SinSR(1) | OSEDiff(1) | Ours(1) |
|---|---|---|---|---|---|---|---|---|---|
| Infer. | 12.4151 | 7.9637 | 16.8704 | 4.8441 | 0.7546 | **0.1416** | 0.1424 | 0.1791 | 0.1791 |
| Para. | 150.0 | 380.0 | 1331.2 | 625.0 | 118.6 | 33.8 | 118.6 | **8.5** | **8.5** |

**Quantitative Comparison**. We use RealESRGAN as a simulation of real-world degradation and compare the performance on the DIV2K, RealSR, and DRealSR validation sets. Tab. 2 and Tab. 3 present the quantitative comparison results.

Tab. 2 compares our method with existing methods that excel in terms of fidelity, showing that our method is comparable in terms of PSNR and SSIM, while significantly outperforming others in realness metrics such as DISTS, FID, and others. The comparison with RealESRGAN further demonstrates that diffusion-based methods generally achieve higher scores on no-reference metrics (NIQE, MANIQA, CLIPIQA, MUSIQ), suggesting that diffusion models are better suited to provide visual priors for super-resolution tasks. Tab. 3 compares our method with existing methods that excel in realness. The results show that our method is competitive in realness metrics while also achieving significant performance gains in fidelity.

**Qualitative Comparison**. Fig. 4 presents the results of comparison experiments on RealSR testset. The figure shows that our method provides better visual quality and consistency with the original image compared to the other methods, proving that our CTSR achieves better image quality, PSNR and DISTS metrics, as well as natural and vivid details. It is also notable that both OSEDiff Wu et al. (2024a), the previous best method, and our CTSR exhibit a hue different from that of other earlier methods, like ResShift Yue et al. (2023), which is possibly due to different color fix settings.

**Efficiency Comparison**. To evaluate the efficiency and complexity of CTSR, we compare these properties with the SOTA methods in Tab. 8, which shows that CTSR requires fewer inference steps, achieves a comparable inference time, and has fewer trainable parameters.

### 4.3 ABLATION STUDY

**Necessity of Teacher Distillation Loss**. A natural question arises: "why do we need two teacher models to achieve the trade-off, given that many methods use $L_2$ loss and LPIPS loss to balance fidelity and realness? " From a theoretical standpoint, the $L_2$-norm, when used as a fidelity constraint, is too sparse and lacks the smoothness necessary to capture the detailed semantic information of the LR input. On the other hand, regularization losses, such as LPIPS, struggle to effectively represent the distribution of natural images. By training SR models on a diffusion prior with various strategies, we can obtain better guidance for balancing fidelity and realness, thereby advancing the Pareto frontier of SR tasks. To further support this, we present results with and without the distillation loss in Tab. 4. The comparison shows that, without the distillation loss, the method reverts to the behavior of earlier GAN-based approaches, achieving better fidelity but suffering a significant decline in realness and visual quality. Since multiple SOTA SR models excel in fidelity performance, to find the best choice for $\mathcal{T}_{fl}$, we also experiment with SinSR Wang et al. (2024b) as the teacher model for dual teacher distillation. The results are presented in Tab. 4.

**Selection of Coefficients $\lambda_{fl}$, $\lambda_{rn}$ and $\gamma_{time}$**. For the balancing coefficients among the loss function terms, we employ a grid search to determine the values that yield the best overall performance. The results of this selection process are shown in Tab. 6.

### 4.4 EVALUATION OF CONTROLLABILITY AND EXTENDABILITY

**Controllability**. Here, we introduce a controllable image super-resolution method enabled by the proposed second stage distillation. Specifically, the controllability of CTSR is determined by the input time step $t$ of the diffusion model, where $t = 0$ corresponds to the best realness and $t = 1$ to fidelity. The input $t$ can be sampled between 0 and 1, allowing user to adjust the balance between these two properties. We evaluate the performance on the DIV2K validation set, with the results presented in Tab. 7. As the input timestep $t$ increases from 0 to 1, fidelity metrics such as PSNR and SSIM improve, while realness metrics like LPIPS begin to decrease. Visual results are shown in Fig. 1(a) and Fig. 7 in **Appendix**.

**Extension to Image Enhancement**. To demonstrate the generalization and versatility of our proposed fidelity-realness distillation method from Sec. 3.3, we extend it to the low-light enhancement (LLE) task, showcasing the performance improvement achieved by this approach. We select two diffusion-based LLE methods: GSAD Hou et al. (2023), which excels in fidelity, and Reti-Diff He et al. (2023), which excels in realness, and apply a training strategy similar to our CTSR. The results, presented in Tab. 5, show that our proposed distillation strategy preserves the fidelity advantage of GSAD while leveraging the model prior from Reti-Diff to enhance realness performance.

## 5 CONCLUSION

This paper proposes CTSR, a distillation-based real-world image super-resolution method that leverages multiple teacher models to strike a trade-off between realness and fidelity. Furthermore, inspired by the working principle of flow matching, to enable controllability between fidelity and realness, this paper explores a controllable trade-off effect by distilling the output distributions of the aforementioned models, enabling a controllable image super-resolution method that is able to be adjusted via input timestep. Experiments on several real-world image super-resolution benchmarks demonstrate the superior performance of CTSR, compared to other competing methods. Additionally, the proposed fidelity-realness distillation approach can be extended to other tasks, such as low-light enhancement, for performance improvement.

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

## A  APPENDIX

In the supplementary materials, we demonstrate additional experimental results, implementation details, discussion, and analysis as follows.

### A.1  PRELIMINARIES

**Diffusion Probabilistic Models** Ho et al. (2020); Song et al. (2021; 2020) are a class of generative models with strong visual prior. The key idea is to model the data distribution by simulating a forward noise-adding process and a reverse denoising process. Let $\mathbf{x}_0$ represent the original image, $\mathbf{x}_t$ be the data at the t-th step of the forward process. The forward process can be described as: $q(\mathbf{x}_t|\mathbf{x}_{t-1}) = \mathcal{N}(\mathbf{x}_t; \sqrt{1-\beta_t}\mathbf{x}_{t-1}, \beta_t\mathbf{I})$, where $\beta_t$ controls the noise added at each step, and $\mathcal{N}(\cdot, \boldsymbol{\mu}, \sigma^2\mathbf{I})$ represents Gaussian distribution with mean $\mu$ and co-variance matrix

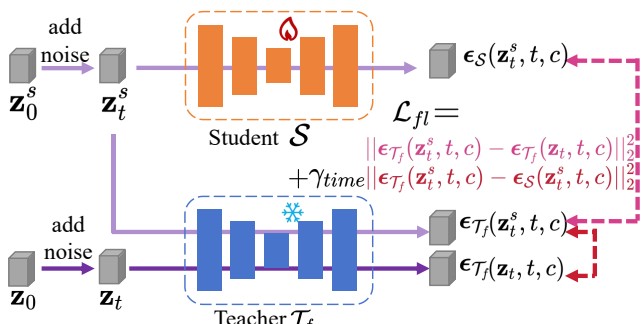

Figure 6: Visualized calculation process of $\mathcal{L}_{fl}$.

$\sigma^2\mathbf{I}$. The reverse process aims to reconstruct the original data $\mathbf{x}_0$ by predicting $\mathbf{x}_{t-1}$ from $\mathbf{x}_t$: $p_\theta(\mathbf{x}_{t-1}|\mathbf{x}_t) = \mathcal{N}(\mathbf{x}_{t-1}; \boldsymbol{\mu}_\theta(\mathbf{x}_t, t), \sigma_t^2\mathbf{I})$, where $\boldsymbol{\mu}_\theta(\mathbf{x}_t, t)$ is the predicted mean parameterized by a neural network.

The training of the diffusion model needs a reconstruction loss of the difference between added noise in forward process, and predicted noise in reverse process, formulated as $L = \sum_{t=1}^{T}[||\boldsymbol{\epsilon}_\theta(\mathbf{x}_t, t) - \boldsymbol{\epsilon}||^2]$, where $\boldsymbol{\epsilon}_\theta(\mathbf{x}_t, t)$ is the model's prediction of the noise $\boldsymbol{\epsilon}$ added at each timestep.

Flow matching Liu et al. (2023); Lipman et al. (2024) is a generative modeling technique similar to diffusion models Meng et al. (2023). It can model and learn the mapping from one data distribution to another through a noise-adding and denoising process, similar to diffusion models. Such distribution transformation process can be applied to tasks such as image reconstruction and style transfer Martin et al. (2024); Dao et al. (2023); Hu et al. (2024); Yin et al. (2024).

**Convex Optimization for Image Restoration** Image restoration, when modeled as $\mathbf{y} = \mathbf{A}\mathbf{x} + \mathbf{n}$, is also known as image inverse problem. The target for image restoration is as $\arg\min_{\mathbf{x}} ||\mathbf{y} - \mathbf{A}\mathbf{x}||_2^2 + \lambda\mathcal{R}(\mathbf{x})$, where $\mathcal{R}(\mathbf{x})$ is the regularization term, like $L_1$ norm or total variation Rudin et al. (1992); Zhu et al. (2024b). This convex optimization problem can be solved via algorithms like gradient descent and ISTA Ito et al. (2019), in an iterative process. Take gradient descent step as an example: $\mathbf{x}_{k+1} = \mathbf{x}_k + \rho\nabla_{\mathbf{x}}(\mathbf{y} - \mathbf{A}\mathbf{x}_k)$, where $\mathbf{x}_k$ and $\mathbf{x}_{k+1}$ is the restoration result in $k$ and $k+1$ step, and $\rho$ is the learning rate. Diffusion-based image SR methods, like DPS Chung et al. (2023) and DDS Hyungjin et al. (2024), are inspired via such process, taking iterative sampling in diffusion as optimization steps.

## A.2 MORE IMPLEMENTATION DETAILS

### A.2.1 MORE DETAILS OF LOSS FUNTION

We provide a detailed loss calculation process for Stage 1 in the main paper, as shown in Fig. 6.

### A.2.2 PSEUDOCODE OF OUR PROPOSED CTSR METHOD

The overall training process for first and second stage is shown in Algo. 1 and Algo. 2.

## A.3 MORE EXPERIMENTAL RESULTS

### A.3.1 MORE ABLATION RESULTS

**Ablation of Stage 2 Distillation** Ablation results for two-stage training are shown below in Tab. 9. Better results in **bold**. It is shown that with second stage of training, our proposed method could also have better performance.

---

**Algorithm 1:** Fidelity-Realness Distillation in Stage 1

---

**Input:** Ground truth $\mathbf{x}_{GT}$, input LR image $\mathbf{x}_{LR}$, student model $\mathcal{S}$, teacher model $\mathcal{T}_{fl}$ and $\mathcal{T}_{rn}$, VAE encoder $\mathcal{E}$, VAE decoder $\mathcal{D}$, embedding of prompt $c$, loss balancing hyper-parameters $\lambda_{time}, \lambda_{fl}, \lambda_{rn}, \lambda_{l2}, \lambda_{lp}$

**Output:** Student model $\mathcal{S}$

1 Initialize $\mathcal{S}$ using weight of $\mathcal{T}_{rn}$.
2 **for** $epoch = 1$ **to** $total\ epochs$ **do**
3    $\mathbf{z}_1 = \mathcal{E}(\mathbf{x}_{LR})$
4    $\mathbf{z}_0 = \mathcal{E}(\mathbf{x}_{GT})$
5    Random sample a timestep $t$
6    $\mathbf{z}_t = add\_noise(\mathbf{z}_0, t)$
7    $\mathbf{z}_0^s = \mathcal{S}(\mathbf{z}_1)$
8    $\mathbf{x}_0 = \mathcal{D}(\mathbf{z}_0^s)$
9    $\mathbf{z}_t^s = add\_noise(\mathbf{z}_0^s, t, c)$
10   $\mathcal{L}_{rec} = \lambda_{l2}||\mathbf{x}_{GT} - \mathbf{x}_0||_2^2 + \lambda_{lp}\ell(\mathbf{x}_{GT}, \mathbf{x}_0)$
11   $\mathcal{L}_{fl} = ||\boldsymbol{\epsilon}_{\mathcal{T}_f}(\mathbf{z}_t^s, t, c) - \boldsymbol{\epsilon}_{\mathcal{S}}(\mathbf{z}_t^s, t, c)||_2^2 + \lambda_{time}||\boldsymbol{\epsilon}_{\mathcal{T}_f}(\mathbf{z}_t, t, c) - \boldsymbol{\epsilon}_{\mathcal{T}_f}(\mathbf{z}_t^s, t, c)||_2^2$
12   $\mathcal{L}_{rn} = ||\boldsymbol{\epsilon}_{\mathcal{T}_r}(\mathbf{z}_t^s, t, c) - \boldsymbol{\epsilon}_{\mathcal{S}}(\mathbf{z}_t^s, t, c)||_2^2 + \lambda_{time}||\boldsymbol{\epsilon}_{\mathcal{T}_r}(\mathbf{z}_t, t, c) - \boldsymbol{\epsilon}_{\mathcal{T}_r}(\mathbf{z}_t^s, t, c)||_2^2$
13   $\mathcal{L}_{s1} = \mathcal{L}_{rec} + \lambda_{fl}\mathcal{L}_{fl} + \lambda_{rn}\mathcal{L}_{rn}$
14   $\mathcal{L}_{s1}.backward()$
15   $\mathcal{S}.update()$
16 **end**
17 **return S**

---

---

**Algorithm 2:** Controllability Distillation in Stage 2

---

**Input:** HR output of student model $\mathbf{x}_0$, student model $\mathcal{S}$, teacher model (weight initalized from student model) $\mathcal{T}_{\mathcal{S}}$, VAE encoder $\mathcal{E}$

**Output:** Student model $\mathcal{S}$

1 **for** $epoch = 1$ **to** $total\ epochs$ **do**
2    Randomly sample timesteps $t$ and $t' \in (0, 1)$ /* ensure $t' > t$ */
3    $\mathbf{z}_t = \mathbf{z}_0 + t\boldsymbol{\epsilon}_{\mathcal{T}_{\mathcal{S}}}(\mathbf{z}_0, t, c)$
4    $\mathbf{z}_{t'} = \mathbf{z}_t + t'\boldsymbol{\epsilon}_{\mathcal{T}_{\mathcal{S}}}(\mathbf{z}_0, t, c)$
5    $\mathcal{L}_{ctrl_{t,t'}} = ||t\boldsymbol{\epsilon}_{\mathcal{T}_{\mathcal{S}}}(\mathbf{z}_t, t, c) - t'\boldsymbol{\epsilon}_{\mathcal{T}_{\mathcal{S}}}(\mathbf{z}_{t'}, t', c) + (\Delta t)\boldsymbol{\epsilon}_{\mathcal{S}}(\mathbf{z}_t, t, c)||_2^2$
6    $\mathcal{L}_{s2} = \sum_{t,t' \in [0,1]} \mathcal{L}_{ctrl_{t,t'}}$
7    $\mathcal{L}_{s2}.backward()$
8    $\mathcal{S}.update()$
9 **end**
10 **return** $\mathcal{S}$

---

Table 9: Ablation of second stage distillation. Best results in red.

| Method | PSNR | SSIM | NIQE | CLIPIQA | MANIQA |
|---|---|---|---|---|---|
| w/o $2^{nd}$ stage | **24.36** | 0.6092 | 3.5732 | 0.6737 | 0.5879 |
| w/ $2^{nd}$ stage (Ours) | 24.34 | **0.6093** | **3.5455** | **0.6741** | **0.5894** |

Table 10: More results of the controllable trade-off between fidelity and realness, with adjustable properties implemented via timestep $t$. Test on the **DIV2K** validation set.

| Timestep $t$ | PSNR↑ | SSIM↑ | LPIPS↓ | DISTS↓ | FID↓ | NIQE↓ | MUSIQ↑ | MANIQA↑ | CLIPIQA↑ |
|---|---|---|---|---|---|---|---|---|---|
| 0.0 | 24.34 | 0.6093 | **0.3377** | **0.1377** | **24.56** | **3.5455** | **69.52** | **0.5894** | **0.6741** |
| 0.2 | 24.45 | 0.6098 | 0.3384 | 0.1394 | 24.75 | 3.6803 | 69.25 | 0.5826 | 0.6726 |
| 0.4 | 24.58 | 0.6131 | 0.3397 | 0.1412 | 25.00 | 3.8114 | 69.00 | 0.5767 | 0.6715 |
| 0.6 | 24.72 | 0.6172 | 0.3409 | 0.1432 | 25.64 | 3.9368 | 68.60 | 0.5698 | 0.6684 |
| 0.8 | 24.82 | 0.6191 | 0.3423 | 0.1447 | 26.13 | 4.0234 | 68.25 | 0.5642 | 0.6632 |
| 1.0 | **24.85** | **0.6192** | 0.3437 | 0.1459 | 26.32 | 4.0438 | 67.96 | 0.5609 | 0.6585 |

### A.3.2 More Results of Controllable Image SR

Here we present the controllable image SR effect on the validation sets of DIV2K, RealSR and DRealSR. Results are shown in Tab. 10, Tab. 11 and Tab. 12 seperately.

### A.3.3 More Visual Results

We provide more results presenting the controllability of our proposed CTSR, which are shown in Fig. 7. From left to right, the fidelity property is gradually changed to realness, with less smooth and more details and better visual quality. We also provide a detailed comparison result between our CTSR and OSEDiff Wu et al. (2024a) in Fig. 5. It is shown that output of our method have more vivid details, like the fine wrinkles and folds on the forehead, as well as the brick textures on the wall.

### A.4 More Related Work and Discussion

### A.4.1 More Related Work on Image SR

**GAN-based and MSE-oriented Image SR Methods** Earlier work mainly use GAN Goodfellow et al. (2014) and MSE-oriented Vaswani et al. (2017); Dong et al. (2015) networks to implement the image SR task Ren et al. (2020); Wang et al. (2021); Pan et al. (2021); Wang et al. (2018); Yinhuai et al. (2023); Poirier-Ginter & Lalonde (2023). SRGAN Ledig et al. (2017) first uses the GAN network to image SR task, optimized via both GAN and perceptual losses, to improve visual quality. Based on this observation, ESRGAN Wang et al. (2018) improved detail recovery by incorporating a relativistic average discriminator. Methods like BSRGAN Zhang et al. (2021a) and Real-ESRGAN Wang et al. (2021) follow the complexities of real-world degradation, allowing the ISR approaches to effectively tackle uncertain degradation, thus improving the flexibility of the model. Although GAN-based methods can inject more realistic detail into images, they struggle

Table 11: More results of the controllable trade-off between fidelity and realness, with adjustable properties implemented via timestep $t$. Test on the **RealSR** testset.

| Timestep $t$ | PSNR↑ | SSIM↑ | LPIPS↓ | DISTS↓ | FID↓ | NIQE↓ | MUSIQ↑ | MANIQA↑ | CLIPIQA↑ |
|---|---|---|---|---|---|---|---|---|---|
| 0.0 | 25.70 | 0.6962 | **0.3058** | **0.1530** | **121.30** | **4.0662** | **67.94** | **0.6367** | **0.6495** |
| 0.2 | 26.29 | 0.7211 | 0.3210 | 0.1620 | 127.67 | 4.2979 | 66.84 | 0.6314 | 0.6435 |
| 0.4 | 26.61 | 0.7203 | 0.3178 | 0.1594 | 134.38 | 4.2320 | 66.33 | 0.6355 | 0.6340 |
| 0.6 | 26.62 | 0.7204 | 0.3191 | 0.1605 | 145.21 | 4.2561 | 65.29 | 0.6340 | 0.6333 |
| 0.8 | 26.65 | 0.7208 | 0.3206 | 0.1614 | 148.86 | 4.2708 | 62.64 | 0.6327 | 0.6240 |
| 1.0 | **26.72** | **0.7213** | 0.3220 | 0.1628 | 156.38 | 4.3209 | 61.08 | 0.6304 | 0.6209 |

Table 12: More results of the controllable trade-off between fidelity and realness, with adjustable properties implemented via timestep $t$. Test on the **DRealSR** testset.

| Timestep $t$ | PSNR↑ | SSIM↑ | LPIPS↓ | DISTS↓ | FID↓ | NIQE↓ | MUSIQ↑ | MANIQA↑ | CLIPIQA↑ |
|---|---|---|---|---|---|---|---|---|---|
| 0.0 | 27.38 | 0.7767 | **0.3423** | **0.1937** | **142.52** | **6.6438** | **64.70** | **0.6412** | **0.7060** |
| 0.2 | 27.53 | 0.7794 | 0.3446 | 0.1402 | 147.25 | 7.7594 | 63.52 | 0.6408 | 0.7042 |
| 0.4 | 27.99 | 0.8023 | 0.3513 | 0.1687 | 150.39 | 7.5088 | 63.35 | 0.5654 | 0.6958 |
| 0.6 | 28.22 | 0.8043 | 0.3528 | 0.2195 | 156.36 | 7.5306 | 62.99 | 0.5642 | 0.6930 |
| 0.8 | 28.47 | 0.8056 | 0.3561 | 0.2369 | 161.24 | 7.8462 | 58.76 | 0.5453 | 0.6745 |
| 1.0 | **28.68** | **0.8152** | 0.3697 | 0.2371 | 164.46 | 7.9699 | 57.85 | 0.5974 | 0.6664 |

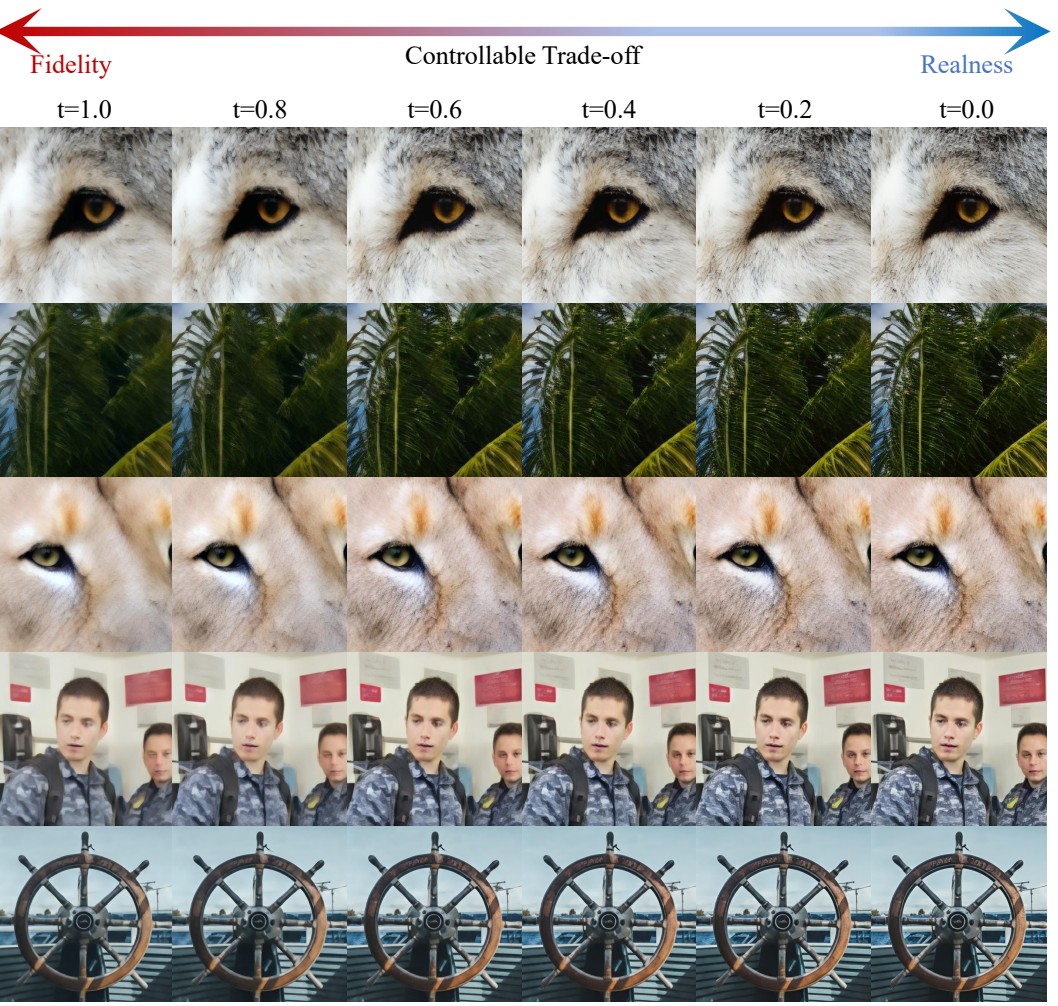

Figure 7: Visualized results of controllable image SR.

with challenges such as training instability. For MSE-oriented methods, SwinIR Liang et al. (2021) introduces a strong baseline model for image restorations, which includes image super-resolution (including known degradation and real-world types), image denoising, and JPEG compression artifacts. As this method is also trained in an end-to-end manner, it also faces problems like over-smooth and detail missing.

### A.4.2 MORE DISCUSSION OF CONTROLLABLE IMAGE SR APPROACHES

Recent works such as PiSA-SR Sun et al. (2025) and OFTSR Zhu et al. (2024c) have explored diffusion-based approaches for real-world image super-resolution (SR), which is controllable between fidelity and realness. Here we discuss the difference between our CTSR and these approaches briefly.

**Comparison with PiSA-SR**. Our proposed method differs from PiSA-SR in both formulation and implementation. Specifically, we adopt a flow-matching training strategy that fine-tunes the initial-stage SR model to establish a continuous mapping within the solution space, from high-fidelity outputs to those with improved perceptual realness. This enables controllable super-resolution by navigating along the learned flow. In contrast, PiSA-SR explicitly decouples fidelity and realness into separate objectives at the pixel and semantic levels, respectively. It fine-tunes two dedicated LoRA modules to address each aspect and leverages different LoRA weightings to balance fidelity and realness. This leads to a fundamentally different control mechanism compared to our continuous and unified flow-based strategy.

**Comparison with OFTSR**. While both OFTSR and our method leverage flow-based models for controllable SR, there are significant differences in both conceptual framework and practical implementation. From the perspective of domain optimal transport via flow matching, OFTSR distills denoising trajectories directly from an initial latent point toward the high-fidelity and high-realness domains. The trajectory direction is implicitly controlled by selecting different timesteps, and the entire distillation process is completed in a single stage. In contrast, our CTSR method decomposes this process into two stages: in the first stage, we obtain a strong SR model via dual-teacher distillation process that simultaneously enhances both fidelity and realness, serving as a unified initial trajectory endpoint aligned with the targets of OFTSR. In the second stage, we further refine the mapping along a constrained subspace, learning a directional flow from fidelity to realness. This staged decomposition provides finer control over the trade-off between fidelity and realness, and reflects a key difference between our approach and OFTSR.

Moreover, OFTSR assumes a known and fixed degradation operator, which limits its applicability to synthetic or well-characterized degradation settings. In contrast, our CTSR framework is designed for real-world SR scenarios, where degradation types and parameters are unknown and potentially diverse. This makes CTSR more suitable for practical applications where the degradation process is complex and not explicitly defined.

### A.4.3 MORE DISCUSSION OF DISTRIBUTION DISTILLATION AND KNOWLEDGE DISTILLATION

Although motivated by the success of knowledge distillation in image SR task, our method differs from these distillation methods in both objective and distillation design. Traditional knowledge distillation methods targets at tasks like classification, via techniques including adaptive multi-teacher fusion and multi-level supervision. Ours distillation is inspired from Score Distribution Sampling Poole et al. (2022) and Variational Score Distillation Wang et al. (2023c); Wu et al. (2024a), which use pre-trained generation prior to regulate the generation process of student model, making the distribution of generated result closer to the distribution of pre-trained generation model. Also, our CTSR tackles real-world image super-resolution instead of typical knowledge distillation task.

### A.5 LLM USAGE DECLARATION

In the preparation of this document, we utilized Large Language Model (LLM) to enhance the quality of the writing. Its application is focused on text polishing, grammar correction, and improving clarity. All content generated with the assistance of the LLM was rigorously reviewed, revised, and ultimately approved by the authors to ensure its accuracy and originality.

