# OpenReview forum: "CTSR: Controllable Fidelity-Realness Trade-off Distillation for Real-World Image Super Resolution"
_ICLR.cc/2026/Conference — ICLR 2026 Conference Withdrawn Submission_

### Official Review · Reviewer_LojG · 2025-10-28

**Soundness:** 3
**Presentation:** 2
**Contribution:** 2
**Rating:** 4
**Confidence:** 5

**Summary:**

This paper proposes CTSR (Controllable Trade-off Super-Resolution), a novel method for real-world image super-resolution that balances fidelity and visual realness. Motivated by the observation that combining multiple models outperforms single ones, the authors use a distillation-based approach to integrate the strengths of different teacher models. CTSR also introduces a controllable mechanism to flexibly adjust the trade-off between fidelity and realness. Experiments on several benchmarks show that CTSR outperforms state-of-the-art methods in both aspects.

**Strengths:**

1. Controllability: Allows flexible adjustment of the fidelity-realness trade-off.

2. Strong Results: Outperforms state-of-the-art methods on multiple real-world benchmarks.

**Weaknesses:**

1. It is unclear how the velocity field (v_S) is learned. Does it require a separate network? Please clarify the learning mechanism.

2. In Table 6, increasing $λ_{fl}$ does not improve PSNR (fidelity), and increasing $λ_{rn}$ does not improve LPIPS (realness). Please explain why.

3. The $γ_{time}$ term in Eq. (2) and (3) seems unrelated to distillation. What is its purpose? What happens when it is set to zero? Please include an ablation study.

4. As shown in Table 7 and Fig. 1, varying t leads to minimal changes in both metrics and visual results. What is the practical value of the controllability? Please provide real use cases or examples.

5. PiSA-SR [1] also proposes a controllable super-resolution method. Please clarify the differences and advantages of CTSR compared to PiSA-SR.

[1] Sun L, Wu R, Ma Z, et al. Pixel-level and semantic-level adjustable super-resolution: A dual-lora approach[C]//Proceedings of the Computer Vision and Pattern Recognition Conference. 2025: 2333-2343.

6. The locations of Table 2 and Table 3 are too far from Section 4, making it hard for readers to locate relevant results quickly. Consider moving them closer to improve readability.

**Questions:**

1. CTSR distills from two teacher models, yet the results surpass both. Please explain why this happens.

2. Choice of Fidelity Teacher – Concerns:
a. Domain Gap with ResShift:
ResShift is not based on the same pretrained diffusion model (SD 2.1-base), and its VAE encoder differs from SD 2.1. Using VSD-type distillation directly may cause a domain mismatch. How is this issue addressed?

b. Why not train a simple fidelity teacher using only L2 loss?

---

### Official Review · Reviewer_fujg · 2025-10-29

**Soundness:** 3
**Presentation:** 3
**Contribution:** 3
**Rating:** 4
**Confidence:** 3

**Summary:**

This paper introduces a novel framework for real-world image super-resolution that enables users to flexibly control the trade-off between fidelity and realness, by distillation of two teachers that excels at fidelity and realness respectively. A flow-matching training is then conducted to obtain a controllable model. The experiments show CTSR demonstrates better fidelity-realness trade-off compared to leading SR methods.

**Strengths:**

1. Distilling two teachers with distinct specialties to obtain a better trade-off is a reasonable design.
2. Developing a model where the fidelity and realness trade-off is controllable is with great research value and real-world usage.
3. The experiments show that, by selecting the right time-step t, the proposed method can achieve significantly better performance than SOTA method.

**Weaknesses:**

1. From the results in Table 2 and 3, I can see that time-step t is a critical variable that controls the performance of the proposed method. different t is selected for evaluation different datasets and for comparison with different types of baseline methods. I think in real-world application, we may need a concrete rules or principles on how to select t for specific requirements. Can the authors provide some analysis on the guidelines and principles to selecting t. How is t selected during the experiments in this paper?
2. The Stage 1 model is explicitly trained to balance fidelity and realness, not to optimize for realness alone. Using its output as the realness endpoint during inference may misrepresent the actual perceptual optimum and could affect the validity of the controllable interpolation claim. It would strengthen the paper if the authors clarified this design choice and provided empirical justification.
3. Figure 2 is a bit confusing and not well explained in the main paper. What space does figure 2 represent? is it the 2-d space of realness and fidelity?

**Questions:**

Please address my concerns in the weaknesses

---

### Official Review · Reviewer_ocvu · 2025-10-29

**Soundness:** 2
**Presentation:** 3
**Contribution:** 2
**Rating:** 4
**Confidence:** 3

**Summary:**

The paper proposes CTSR, a controllable real-world image super-resolution (SR) framework that explicitly balances fidelity and realness. The method proceeds in two stages. First, it distills knowledge from two teacher models: a high-fidelity SR model and a high-realness SR model, which guide the student model to aware pareto front. Second, it learns a continuous transformation that enables controllable traversal along the fidelity realness trade-off at inference time using the control parameter. Experiments on several benchmarks show that CTSR outperforms recent diffusion-based approaches.

**Strengths:**

1.	The paper clearly defines the fidelity-realness trade-off as a multi-objective optimization problem, effectively motivating the need for a distillation approach over naive model interpolation.

2.	The method is technically novel, employing a dual-teacher distillation for complementary guidance, followed by a flow-matching-inspired approach to achieve continuous controllability.

3.	The paper conducts extensive experiments on multiple real-world SR benchmarks, demonstrating competitive performance against recent methods while offering the significant advantage of single-step inference.

**Weaknesses:**

1.	The paper does not make a crisp argument for what is fundamentally new versus (i) multi-teacher knowledge distillation for SR and (ii) interpolating between latent codes plus flow matching based transport. It seems like CTSR is a combination of existing methods. I recommend adding a detailed comparison with: controllable SR via dual LoRA/adjustable guidance strength, flow-matching approaches and prior multi-teacher SR distillation.

2.	The loss function in Stage 1 is a combination of reconstruction, dual-teacher distillation, and a latent-space consistency term. The paper claims that the consistency term (weighted by $γ_{time}$) is superior to a standard L2 loss for preventing oversmoothing (line 241), but this central design choice lacks direct empirical validation. The current ablation studies (Tab. 4) only consider "with/without distill" and "teacher", but do not isolate the contribution of individual components within the proposed loss, e.g., setting $γ_{time} = 0$.

3.	The paper says it targets real-world SR, but currently most experiments simulate degradation with RealESRGAN’s operator and then evaluate on DIV2K/RealSR/DRealSR after ×4 upsampling. This pipeline is common, but I think it is still synthetic real-world style rather than true LR photos with unknown operations like downsampling and compression. I wonder if it is possible to provide any results that are on native RealSR LR photos without synthetic degradation injected.

**Questions:**

1. In Stage 2, you define $\pi_0$ = outputs of the distilled Stage 1 student ($S_1$) and ($\pi_1$) = outputs of the high-fidelity teacher ($T_f$), then train a field $v_S$ in latent space to approximate straight-line transport between them. At inference you effectively take one Euler step from $z_0$ in the predicted direction, scaled by t. First, does this imply that all intermediate outputs lie on (or near) a single linear path between these two endpoints, regardless of image content? If so, is the controllable frontier actually one-dimensional for every input? What’s more, how do you prevent runaway hallucinations when t>1 or t<0 in your experiment?

2. At inference, do you require the teachers at all, or is the final controllable student fully standalone?

---

### Official Review · Reviewer_XEwN · 2025-11-01

**Soundness:** 3
**Presentation:** 3
**Contribution:** 3
**Rating:** 4
**Confidence:** 5

**Summary:**

This paper proposes CTSR, a controllable super-resolution method that balances fidelity and realness. It uses a two-stage training process: first, a dual-teacher distillation guides the model to learn a balanced output; then, a flow matching technique enables smooth control along the fidelity-realness trade-off.

**Strengths:**

- Clear motivation addressing the real-world need to balance fidelity and realism in SR.
- Well-designed two-stage training combining dual-teacher distillation and flow-based control.
- Controllable output with a single parameter, offering flexibility for different use cases.

**Weaknesses:**

- Pareto trade-off claim lacks strong theoretical or visual evidence (e.g., no clear P-D frontier plot).
- Controllability evaluation is limited — few intermediate points shown, no user study or perceptual analysis.
- Teacher model choice is fixed — no analysis of how different teachers impact performance.
- Training complexity is relatively high due to two-stage design and reliance on pretrained teachers.
- OSEDiff is a one-step diffusion model, while the VSD framework typically relies on multi-step diffusion processes to compute meaningful gradient guidance across time steps.
- ResShift and OseDiff operate in different architectures and latent spaces, how do you handle the latent space mismatch between these two teachers?

If this concern is properly addressed, it would improve the overall score.

**Questions:**

see the weaknesses

---

### Note · Authors · 2025-11-14

I have read and agree with the venue's withdrawal policy on behalf of myself and my co-authors.